# A Unified Pervasive Linebroadening Function for Quantum Wells in Light Emitting Diodes

**Juha Viljanen**

Consulting Photonics Scientist, FIN-02180 Espoo, Finland; juha.viljanen@iki.fi; Tel.: +358-405270027



**Featured Application: Linebroadening functions are essential in extracting material quality data from the optical spectra. The presented, unified lineshape function applies to the simulation of inhomogeneously and homogeneously broadened LED lineshapes.**

**Abstract:** The broadening functions for quantum wells in LEDs and laser diodes below the lasing threshold are examined. Inhomogeneous and homogeneous broadening mechanisms are included. Hydrogen-atom-like exciton and the electron-hole plasma recombination models are considered. Material disorder and the Urbach tail are reviewed as the main reasons for the inhomogeneous broadening. Charge carrier scattering and relaxation times in the conduction and valence bands are examined as the origin for the homogeneous lifetime broadening. Two homogeneous lineshapes are compared using the momentum relaxation times obtained from the electron and hole mobilities available for GaAs. In addition to crystal disorder, the mutual collision of charge carriers in the quantum wells is examined as the reason for the relaxation time shortening. The analogy to pressure broadening in gases is used to combine the proper homogeneous and inhomogeneous broadening functions to a unified quantum well lineshape.

**Keywords:** light emitting diodes; laser diodes; spectral broadening; lineshape functions; quantum wells; spectral simulations

## 1. Introduction

Quantum well (qw) LEDs and laser diodes are the primary photon generators in numerous modern lighting and photonics devices. For these applications, it is crucial to be familiar with the qw-spectra and the related physics. The wavelength of the spectral intensity maximum is controlled by adjusting the dimensions and material composition in the quantum well. However, for the spectral shape, it is necessary to know also the physical broadening mechanisms and their characteristic functions. They are examined here and compared against the spectral broadening in gases. This resemblance is useful for reviewing the homogeneous and inhomogeneous broadening effects and postulating a unified and pervasive qw-lineshape. Pressure-related lifetime broadening [1,2] is used to describe effects arising from the increased carrier density in the quantum well confinement. Physically relevant lineshape functions are convolved with the theoretically predicted unbroadened emission spectrum. Convolution theorem and fast Fourier transform (FFT) are applied to evaluate the convolution integrals [3]. The resulting lineshapes are compared against each other and to the experimental data from the literature. The postulated unified broadening function is concluded to be valid for both the homogeneously and inhomogeneously broadened qw-LED spectra.

Optical emission and absorption lines are classified to be homogeneous if all active population members behave identically, which requires they have identical transition energies and probabilities. In gases, the active population consists of freely moving atoms or molecules. Their velocities follow the Maxwell–Boltzmann distribution determined by the temperature. The related Doppler-shifts disperse

the population and make the transition energies inhomogeneously broadened. In normal conditions and low gas pressures, the Doppler broadened, Gaussian distribution represents the minimum line width at a given temperature:

$$L_{ih}(\Delta E) = \frac{1}{\sigma\sqrt{2\pi}}\exp(-\frac{\Delta E^2}{2\sigma^2}). \tag{1}$$

Here $\Delta E$ is the energy deviation from the unbroadened spectral line and $\sigma$ the line width parameter. With increasing pressure, the gas molecules collide more frequently with each other. Collisions decrease the lifetime of the optical energy levels and make them lifetime broadened. The average collision frequency depends on the gas molecule size, pressure, and temperature. Through lifetime broadening, it determines the average linewidth of the population members. At high enough pressures, the originally Gaussian lineshape gradually transforms to a Lorentzian line typical for the homogeneous broadening:

$$L_h(\Delta E) = \frac{\tau/(\pi\hbar)}{1 + \tau^2\Delta E^2/\hbar^2}, \tag{2}$$

where $\tau$ is the lifetime parameter. Between the two extremes of Equations (1) and (2) the spectrum follows the Voigt profile [2], obtained by convolving these two functions. The associativity of the convolution integral guarantees that the convolution with the Voigt function produces identical results as would be obtained by convolving with the Gaussian and Lorentzian functions one after another.

In LEDs, the emission originates from the electron and hole pair recombinations, and the broadening is at the easiest observable just below the bandgap energy. Instead of perfect and abrupt bands, in measurements, semiconductors usually show more or less exponential band edges. Exponential Urbach tail is a usual absorption edge behavior in disordered solids and often met also in the emission and absorption spectra of semiconductor materials [4–8]. It is also common in the electroluminescent (EL) and photoluminescent (PL) spectra of quantum well LEDs [9–12]. The Urbach effect reflects the tail in the joint density of states (JDOS), combining the broadening in the valence and conduction band energy states. Instead of the Doppler effect, it arises from the lattice potential disorder inside the material. We examine only the simplest two lineshapes, that in convolution with the unbroadened bulk, and quantum well JDOS functions give the Urbach tails. They are based on Gaussian disorder distributions. However, also other lineshapes convolving into exponential band tails are available in the literature [13,14].

Also, in quantum wells, spectral broadening phenomena are either inhomogeneous or homogeneous. In suitable conditions, both components can be identified and follow the pressure broadening like behavior. With increasing diode current, the originally inhomogeneous qw-spectrum deviates from the exponential shape and broadens into the bandgap. In this case, a unified pervasive (Voigt function like) lineshape is needed [13], but the describing functions are not the ones given in Equations (1) and (2).

Two models are examined for the e-h pair recombination and lifetime broadening. One is based on the annihilation of hydrogen-atom-like excitons, a paradigm commonly applied with the Lorentzian line shape and a single relaxation time parameter. The other model is the direct recombination within the e-h pair plasma, between the conduction and valence bands. It is more demanding since also the k-vector conservation has to be considered while the emitted photons carry only small momentum [15]. In this case, the resulting lineshape develops into the product of two Lorentzian lines [9]. These two recombination models are compared against the GaAs Urbach tails [4–6], using the electron and hole relaxation times obtained from electrical mobility measurements and calculations.

Figure 1 shows the four examined broadening functions plotted both on linear ($E > 1.4$) and logarithmic scales ($E < 1.4$). On the right-hand side (the linear scale), it is seen that with appropriate parameter selections, three of the four curves can be made to overlap each other down to roughly half of the peak intensity. For a reliable spectral analysis and conclusions, we should therefore, examine the resulting spectral behavior at least on one full log10 scale intensity range.

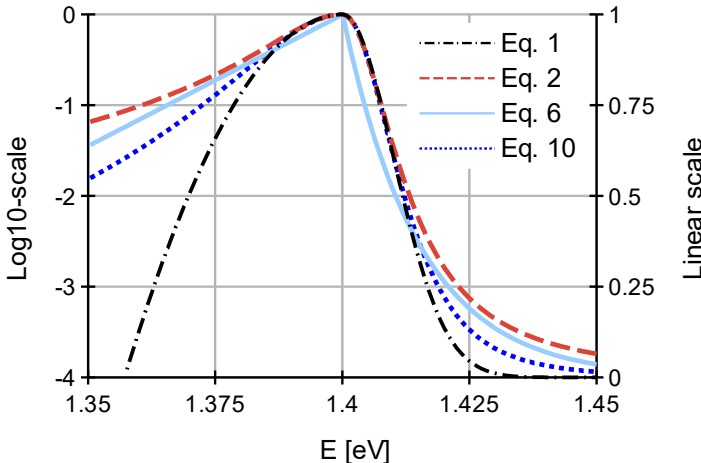

**Figure 1.** Comparison of four lineshapes: Equation (1): ($\sigma$ = 15 meV), Equation (2): ($\tau$ = 50 fs), Equation (6): ($E_u$ = 15 meV), Equation (10): ($\tau_{be} = \tau_{bh}$ = 35 fs).

## 2. The Unbroadened Spectrum

Using Fermi's golden rule the unbroadened spontaneous emission spectrum is obtained from [16]:

$$R_{sp}(E) = \frac{ne^2}{\epsilon_0 \hbar^2 \pi c^3 m_0^2} |M_T|^2 \times E \cdot \rho_{Jdos}(E) \cdot S(E) \cdot f_c(E) \cdot (1 - f_v(E)), \tag{3}$$

where $E$ is the photon energy, $n$ the refractive index, $|M_T|^2$ the squared transition matrix element and $\rho_{Jdos}(E)$ the unbroadened joint density of states (JDOS). $S(E)$ is the Sommerfeld enhancement for the exciton effects, and $f_c$ and $f_v$ are the Fermi-Dirac occupancy functions for the conduction band electrons and valence band holes, respectively. For quantum well LEDs, $\rho_{Jdos}(E)$ follows the Heaviside step function $H(E_0)$ [17], where $E_0$ is the difference between the lowest and highest energy levels in the conduction and valence band quantum wells, respectively. Without significant error, the product $f_c \cdot (1 - f_v)$ can be approximated with the Boltzmann distribution [15], ($\text{Exp}(-E/k_B T)$) and members before the multiplication sign ($\times$) assumed to be constant or slowly varying throughout the studied energy range.

In bulk materials, the exciton continuum enhancement for the oscillator strengths is given by the Elliot formula [18]:

$$S(E) = \pi\alpha \frac{e^{\pi\alpha}}{sinh(\pi\alpha)}, \tag{4}$$

where $\alpha = \sqrt{E_b/(E - E_g)}$ and $E_b$ and $E_g$ are the exciton binding and bandgap energies respectively. For quantum wells, the exciton enhancement $S(E)$ depends on the polarization and propagation directions of the emitted photons. However, for the most common direction, perpendicular to the quantum well plane it is necessary to consider only the TE-polarization, i.e., electric field vector parallel with the qw-plane.

A similar analytical oscillator strength formula is not at hand for quantum wells in diode pn-junctions (i.e., areas with high electric field). Therefore, the available enhancement factor for two-dimensional excitons [19,20] is now applied as an approximation. At usual ambient temperatures, we can neglect the discrete hydrogen-atom-like exciton peaks and the related exciton quasicontinuum in the bandgap. Then we get the 2-D exciton true continuum TE-polarization enhancement from:

$$S_{TE}(E) = \frac{e^{\pi\alpha}}{cosh(\pi\alpha)}. \tag{5}$$

$E_b$ is typically bigger in quantum wells than in bulk semiconductors. Values between 6 and 15 meV have earlier been used for GaAs and related materials [21]. It has been commented that Equation (5) exaggerates the exciton effect in real quantum wells [22]. However, it does not make a significant error here since Equation (5) only varies between 2 and 1, when E increases from $E_g$ to $\infty$ and especially if we examine only the low energy side broadening and the lineshape in the bandgap.

## 3. Inhomogeneous Broadening

### 3.1. The Urbach Line Shape

In bulk solid materials, the exponential low energy spectral tail may span several orders of magnitude down in the bandgap. It arises from the disorder in the periodic crystal potential [23,24] that is either frozen in the material during the fabrication or created by phonon vibrations at the instance of the e-h pair recombination or creation in the optical emission or absorption, respectively. Therefore, at usual ambient temperatures, Urbach tails can exist even in perfect crystals. It has been shown that with a Gaussian disorder depth distribution, the original (disorder free) band states disperse in energy with an exponentially decreasing deviation probability. At the same time, the original abrupt band edges disperse outlined by the convolution with this probability distribution. Since the disorder originates from local variations, the emerging DOS tails are classified as inhomogeneous and are correlated in the conduction and valence bands (disorder related charge carrier scattering is examined in Section 4).

Equation (3) includes the abrupt JDOS, and two convolution operations would be needed to add the conduction and valence band tails separately. A detailed analysis of the broadened JDOS should also include the oscillator strengths in these band tail states [24] and the contribution of excitonic states in the bandgap. However, exciton peaks are absent in the LED spectra, and Urbach tails recorded on bulk semiconductor materials at usual ambient temperatures. Starting from the abrupt JDOS, we can, at the moment, reach only semiempirical broadening functions to combine the conduction and valence band tails. This does not turn down their value in the disorder analysis with experimental absorption and emission spectra.

Equations (6) and (7) are the simplest lineshapes that in convolution with the original bulk material and quantum well $\rho_{Jdos}(E)$ functions produce the observed exponential tails. A detailed microscopic analysis would be needed to study the spectral symmetry of the JDOS broadening and the correlations between the conduction and valence band tail states. Without it, we can examine the symmetry effects by comparing the fully symmetric, Equation (6), and maximally asymmetric, Equation (7), Urbach tail functions:

$$L_{Jdos}(E) = \exp[-|E - \epsilon|/E_u].$$

(6)

$$L_{Jdos}(E) = \frac{1}{2}(1 - sgn(E - \epsilon)) \cdot \exp[-(E - \epsilon)/E_u].$$

(7)

Here $E - \epsilon$ is the deviation from the unbroadened band state energy $\epsilon$, and $E_u$ is the Urbach broadening parameter. In convolution integrals, both functions produce the requested exponential Urbach tails in the bandgap. However, the asymmetric lineshape of Equation (7) produces abnormally sharp-peaked emission lines, as is seen in Figure 2. This leads to the conclusion that the Urbach broadening in the JDOS is more or less symmetric and only Equation (6) is used in the further analysis, though the sharp peak somewhat softens after including the lifetime broadening examined in Section 4.

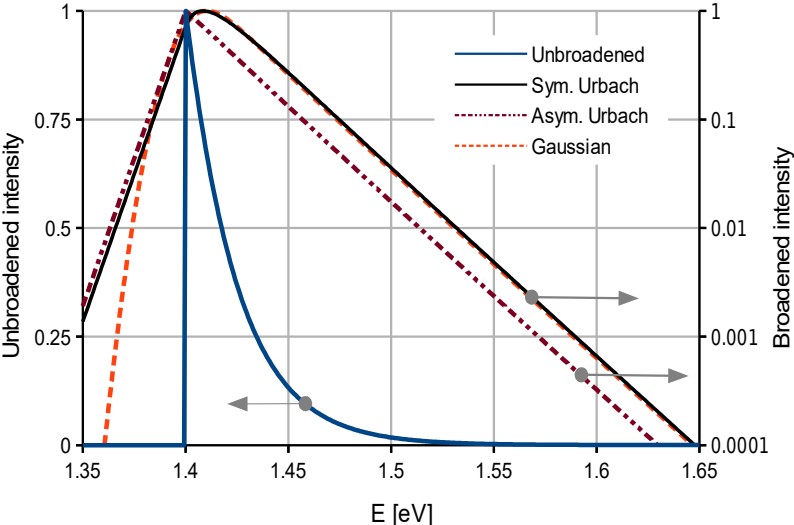

**Figure 2.** Inhomogeneos lineshapes for the unbroadened qw-spectrum, $\epsilon\rho_{Jdos}(\epsilon)S(\epsilon)exp(-\epsilon/k_BT)$, obtained by convolving with three $L_{Jdos}$ lineshapes. In the Urbach functions $E_u$ = 8 meV and in the Gaussian $\sigma$ = 10 meV. [T = 293 K, $E_b$ = 8 meV].

### 3.2. The Inhomogeneously Broadened qw-LED Spectrum

To obtain the inhomogeneous line from the source function in Equation (3) it is convolved with the Urbach $L_{Jdos}$ in Equations (6) or (7).

$$R_{ih}(E) \propto \int_0^\infty \epsilon\rho_{Jdos}(\epsilon)S(\epsilon)exp(-\epsilon/k_BT) \times L_{Jdos}(E - \epsilon)\mathrm{d}\epsilon, \tag{8}$$

Equation (8) can be evaluated either by direct numerical integration or by using the convolution theorem and the fast Fourier transformation (FFT) [3]. The latter approach is preferred here since it is more effective, and another convolution integral will be needed to include also the effect of homogeneous lifetime broadening.

In Figure 2, the asymmetric Urbach function accurately returns the Urbach tail in the bandgap ($E$ < 1.4 eV), but results in an abnormally sharp-peaked emission spectrum. Also, the spectral width narrows, but the shape on the high energy side ($E$ > 1.4 eV) is not affected by the symmetry of the $L_{Jdos}$ function. For comparison, also the Gaussian function in Equation (1) was used as the lineshape, though it does not produce the desired Urbach tail below 1.4 eV like Equations (6) or (7).

### 4. Homogeneous Broadening

Crystal disorder was above considered as the origin of inhomogeneous optical broadening from the JDOS tail. However, it also generates scattering in the conduction and valence bands and causes lifetime broadening in the band states. As the Fourier transform for exponential time decays, Lorentzian lineshape is the obvious and natural lifetime broadening function. With equal scattering cross sections for charge carriers in similar energy levels this broadening process is also homogeneous.

In qw-LEDs, e-h pairs recombine typically between the lowest conduction band and the highest valence band quantum well energy levels, respectively. The emitted photon energy is somewhat smaller than the bandgap in the surrounding materials, which reduces the photon reabsorption. The qw concentrates the charge carriers to a dense e-h plasma, effectively increasing the recombination and photon creation rates. Mutual scattering of electrons and holes [25] is one more process enhanced in the qw. The enhanced recombination and scattering rates decrease the relaxation times in the conduction and valence bands, making the spectral shape dependent on the diode current. At high enough carrier densities, like in the pressure broadening in gases, lifetime broadening buries the

underlying inhomogeneous line shape. Without the qw-confinement, this situation is difficult to reach in bulk semiconductors [9,26] since also the inhomogeneous broadening increases with lattice disorder.

### 4.1. Homogeneus Line Shapes

For direct bandgap transitions without the k-vector conservation constraint in the broadening, the lineshape is obtained by convolving the conduction and valence band Lorentzians [16]. Two Lorentzian functions convolve into a new Lorentzian line with $\tau$ equaling the reduced relaxation time $\tau_r$ (obtained also by Matthiessen's rule for the summing of two scattering frequencies)

$$\tau_r = \tau_e \tau_h / (\tau_e + \tau_h), \tag{9}$$

where $\tau_e$ and $\tau_h$ are the original electron and hole relaxation times in the conduction and valence bands, respectively.

Taking the k-vector conservation into account also in the broadening, the homogeneous lineshape for interband transitions between two Lorentzian broadened band states is [9]:

$$L_h(\Delta E) = \frac{\tau_{be} + \tau_{bh}}{\pi \hbar} \times \frac{1}{1 + \tau_{be}^2 \Delta E^2 / \hbar^2} \times \frac{1}{1 + \tau_{bh}^2 \Delta E^2 / \hbar^2}, \tag{10}$$

where

$$\tau_{be} = \tau_e \cdot m_r / m_e, \quad \tau_{bh} = \tau_h \cdot m_r / m_e \tag{11}$$

are the effective relaxation times and $m_e$ and $m_h$ the effective electron and hole masses, respectively and $m_r = m_e m_h / (m_e + m_h)$. Tensile or compressive quantum well strains are often applied to remove the light-hole heavy-hole band degeneracy [17]. Strain also makes the effective mass ratios solitary and unique for Equation (10).

In Figure 3, the combined inhomogeneous and homogeneous broadenings are plotted with four different $\tau_{be}$ and $\tau_{bh}$ ratios in Equation (10). The inhomogeneous lineshape in the convolution was the Urbach broadened spectrum, where the JDOS follows the Heaviside step function at $E_0$, and the exciton enhancement is included using Equation (5).

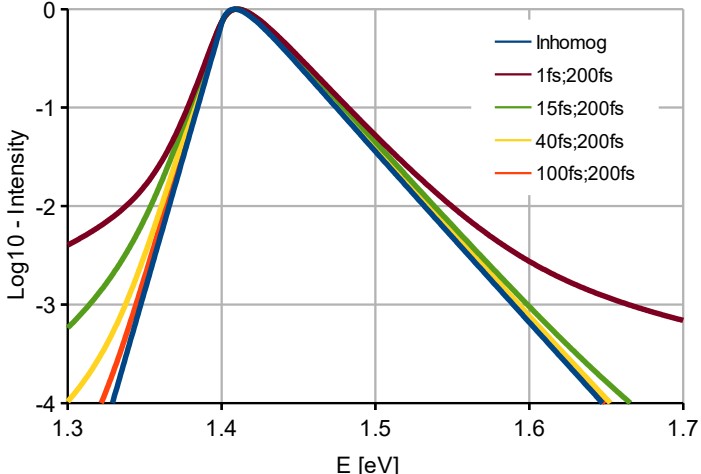

**Figure 3.** Simulation for the homogeneous broadening in a qw-LED, using Equation (10) and inhomogeneous line according to Equation (8) with parameters: $\rho_{Jdos} \propto H(E_0), E_0 = 1.4\,\text{eV}, E_b = E_u = 8\,\text{meV}$ and T = 293 K.

Equation ([10]) can also be written as the difference of two Lorentzian functions [9] revealing that for large relaxation time ratios, it converges to the simple Lorentzian lineshape with the larger relaxation time ($\tau_{be}$ or $\tau_{bh}$) as the linewidth parameter. It additionally obeys the momentum conservation rule in the broadening, and is, therefore, applied as the default homogeneous lineshape. The slimmest lines are obtained with $\tau_{be}$, being equal with $\tau_{bh}$.

### 4.2. Compliance Check against Experimental Spectra

Both the JDOS softening and the charge carrier scattering originate from the lattice disorder, but the latter is very seldom included in the studies for optical spectra. However, these two effects are simultaneously present at all times, and the lifetime broadening exists even in spectra where the exponential Urbach tail spans several orders of intensity magnitude. Likewise, the softening of band edges is usually omitted in the carrier mobility studies in semiconductor materials.

This demands for a compliance check using the momentum relaxation times available from charge carrier mobilities and scattering rate calculations. For gallium arsenide at 300 K, the mobilities are given as: $\mu_e$ = 8500 and $\mu_h$ = 400 cm$^2$/(Vs) for the electrons and holes, respectively (light holes $\mu_{lh}$ = 1067 and heavy holes $\mu_{hh}$ = 341) [27–31]. Using these values and the electron (0.063) and heavy hole (0.51) effective masses [32] for the $m_{eff}$ in Equation ([12]) we get the first line in Table [1], where the heavy hole mass was applied for its higher density of states.

$$\tau = \mu * m_{eff}/e \tag{12}$$

At room temperature in GaAs, the relaxation times are determined mainly by the scattering from longitudinal optical (LO) phonons, with phonon energy, $\hbar\omega_{LO} \approx 35$ meV [27–31]. Up to this energy, LO-phonon absorption dominates the relaxation for the electrons and holes above the conduction band minimum and below the valence band maximum, respectively . The corresponding scattering times are given on the second line in Table [1]. Above the $\hbar\omega_{LO}$ energy, instead of phonon absorption, hot carrier scattering arises from the emission of optical phonons and the associated relaxation times are on the last table line.

**Table 1.** Electron and hole relaxation times [fs] in GaAs at 300 K. The $\tau_r$, $\tau_{be}$ and $\tau_{bh}$ columns were obtained from Equations ([9]) and ([11]) and used in Equations ([2]) and ([10]).

| E(e), E(h) | $\tau_e$ | $\tau_h$ | $\tau_r$ | $\tau_{be}$ | $\tau_{bh}$ |
|---|---|---|---|---|---|
| **from mobility** | 304 | 116 | 84 | 271 | 13 |
| **< $\hbar\omega_{LO}$** | 600 | 260 | 181 | 540 | 27 |
| **> $\hbar\omega_{LO}$** | 168 | 60 | 44 | 150 | 7 |

Finally, to validate these values, the corresponding Lorentzian lineshapes ($\tau_r$ = 84 fs, 181 fs) are convolved with the Inhomog  Urbach spectrum in Figure [4]. It was evaluated that, in order to comply with the Urbach tails, for two or more intensity magnitude orders, [4,5] approximately 2000 fs values would be needed for the $\tau_r$. We conclude that the basic Lorentzian lineshape poorly fits the experimental emission and absorption spectra. It is also problematic for the LED quantum wells [33,34].

On the other hand, using the above $\tau_e$ and $\tau_h$ values in the product of two Lorentzians broadening function in Equation ([10]) it convolves into the plotted lineshapes L (271 fs) $*$ L (13 fs) and L (540 fs) $*$ L (27 fs) and the latter does not significantly diverge from the original Inhomog spectrum.

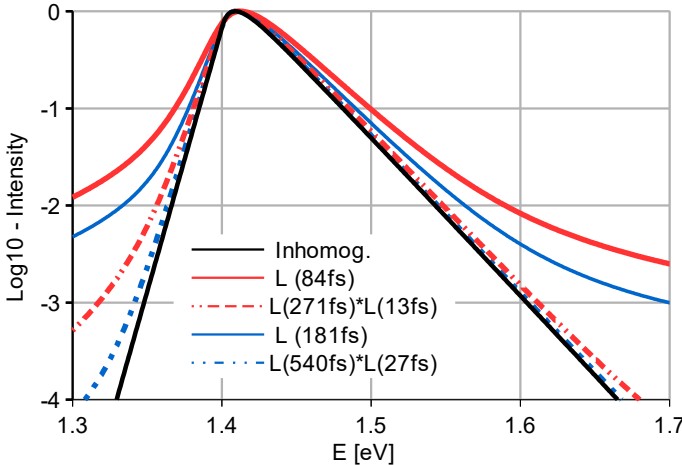

**Figure 4.** Simulation for homogeneous broadening in bulk semiconductor using two homogeneous lineshapes, Equations (2) and (10). Inhomogeneous line with parameters: Bulk $\rho_{Jdos} \propto \sqrt{E - E_g}, E_g = 1.4\,\text{eV}, E_b = 8\,\text{meV}, E_u = 8\,\text{meV}$ and T = 293 K.

## 5. Unified Pervasive Lineshape for Quantum Wells

The associativity of the convolution integral allowed the combination of the Lorentzian and Gaussian lines to a unified broadening function for the pressure broadening in gases [1,2]. Similarly, we may combine the inhomogeneous and homogeneous broadening functions to a single and unified lineshape [13] for the LED quantum wells.

$$L_{LED}(\Delta E) \propto \int_0^\infty L_{Jdos}(\epsilon) L_h(\Delta E - \epsilon) d\epsilon. \tag{13}$$

In this work, the inhomogeneous and homogeneous broadenings follow Equations (6) and (10), respectively. Unfortunately, like the Voigt function, neither can Equation (13) be written in elementary functions. Therefore, instead of applying the closed-form and exponential integral functions, it is easier to work with the convolution theorem and the fast Fourier transform algorithm (FFT).

## 6. Conclusions

Equation (13) is inevitable if the homogeneous and inhomogeneous broadening phenomena are examined simultaneously. In semiconductors with long Urbach tails, the homogeneous contribution can often be neglected. Without a prior assumption on the prevailing broadening mechanism, it is difficult to identify and select the inhomogeneous lineshape function for the observed spectra [35]. Here, the simple exponential function was justified by assuming Gaussian lattice disorder distributions. However, the symmetry of the Urbach line could not be established. The symmetry and parameters $E_0$, $T$, and $E_b$ affect the emission peak wavelength and the spectral width. The lattice disorder related lifetime broadening softens the eventual sharp peak arising from Urbach asymmetry. The spectrally symmetric hyperbolic *Sech* function lineshape also convolves into exponential Urbach tails, but sofar lacks physical explanation.

Three parameters $E_u$, $\tau_{be}$, $\tau_{bh}$ are required in Equation (13) to determine the spectral shape on the bandgap side since only the exciton true continuum was included in Equation (8). This option is justified, while in LEDs, the quantum well typically locates in the high internal electric field of the pn-junction. Separate exciton peaks did neither appear in the photoluminescent (PL) spectra, recorded at usual ambient temperature, though the pn-junction electric field had been compensated by the photovoltage in the diode [9]. At room temperature, the average thermal energy (3/2 kT ≈ 26 meV) is more than twice the typical exciton binding energy, and 300 fs exciton ionization times have been

evaluated in GaAs quantum wells without the pn-junction electric field [36]. If hydrogen-atom-like excitons do not exist long enough to appear in the Urbach tail, Equations (8) and (13) are useful also for the PL-measurements.

qw-LEDs and LDs are fabricated by depositing several semiconductor layers on top of each other. Layer thicknesses are typically close to or only a fraction of the emitted wavelength. In LDs, the quantum well typically locates in the waveguide at the center of the thin-film stack of several high refractive index materials. There, refractive index differences may produce significant optical interference if the spectrum is recorded perpendicular to the quantum well. The magnitude of the interference effects can be evaluated using commercial thin-film optics software [37]. Interference-free PL and EL spectra can be recorded only in the diode junction and qw-plane directions, and with simple optics, it is also possible to separate the TE and TM polarizations in the spectrum [9].

**Funding:** This research received no external funding.

**Conflicts of Interest:** The author declares no conflict of interest.

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
