# Peer review of "A Unified Pervasive Linebroadening Function for Quantum Wells in Light Emitting Diodes"

_applsci, doi:10.3390/app10113774_

Round 1

Reviewer 1 Report

1. It would be good if the author write-down the applications of the study. 2. The sequence of the paper's results need to be more obvious i.e. the starting point of the results show and the end point. 3. the title of number (6. discussion) has to change to( 6. conclusions), also it can be write-down in steps.

Author Response

English language and style:
Reviewer has suggested for spell checking:
After careful and concentrated read through my manuscript, I found two typing errors on the second line on page 7 (row no:164) '...ratio, it coverges to ..'  was corrected to '...ratio, it converges to....'.  Additionally, on the same page on the third line  (line no: 165) '... obeys the moment conservation...' was corrected to '...obeys the momentum conservation ...

Does the introduction provide sufficient background and include all relevant references? The reviewer comments that it can be improved.  
I agree that the literature references given in the introduction are not very recent. The reason is mainly the fact that quantum wells have not anymore for years been a hot research topic. In the recent journals, the focus has more or less shifted from them to quantum dots. Therefore, even the title of this manuscript might be despised as old-fashioned by many modern researchers. Personally I think that even the word 'quantum well' is an old-fashioned misnomer that should in modern journals be changed to something like 'quantum trench' which would describe its shape much better.  

Is the research design appropriate? The reviewer comments that it can be improved.
I agree with this comment. However, within the given five improvement days, it is now, not possible to obtain a significant development. Though this manuscript is far from perfect, I remember my colleague's phrase about academic writing: "Perfect is the worst enemy of good". 

Are the results clearly presented? The reviewer comments that it can be improved.
This manuscript has the nature of a  literature survey.
The point was to dig out the linebroadening functions with physical explanations and backgrounds. Then, two candidates were picked for the inhomogeneous and the homogeneous broadening phenomena.
The selection between the symmetric and asymmetric Urbach lines is vague and based on Fig. 2., where the totally asymmetric function is rejected for its sharply peaked spectrum. However, it was not proven that the inhomogeneous lines are totally symmetric Urbach functions. This statement is included in the last (the 6th) section.  Should this be emphasized more and earlier?
The selection between the two homogeneous linefunctions was clearer and is presented in Fig. 4. Hopefully, there is no need for improvement in that?

However, the main result in this manuscript is the combination of these two selected functions in section 5 in Eq.13. Its idea is to be pervasive by covering both the inhomogeneous and homogeneous broadening effects. The presence of the convolution integral in it is obvious, but the selected functions are unique for this manuscript. In the manuscript, this has been expressed by including reference [4] before introducing Eq.13. Should this be emphasized more? 

Are the conclusions supported by the results? 
The reviewer comments that it can be improved.
For the literature review part
, one should ask instead if the author picked the correct references and functions for the examination.
With the homogeneous broadening, the answer is very likely yes, since no other physically explained lineshapes could be found.
With the inhomogeneous lines, the selection was based on the assumption of Gaussian disorder in the crystal potential and the commonly observed Urbach tail behavior of optical spectra in semiconductor materials. Unfortunately, the symmetry of the Urbach function could not be established. However, the spectral broadening in semiconductors is most easily observed in the bandgap of the materials. There (in Fig. 2) the symmetric and asymmetric Urbach lines produce similar responses and would give the same Urbach parameter, Eu, which is related to the crystal disorder.
At this point, I regard it as very difficult to improve the knowledge of the Urbach line symmetry by searching the literature.
The selections between the two lineshape candidates are based on convolution integrals and graphed in Fig.2 and Fig.4. The selections are also discussed below these figures.

In the comments and suggestions for the authors 
The reviewer states that the author should list the possible applications of this study. 
1) Unfortunately, I am not capable of accomplishing such a list. Obviously the lineshape functions anyhow have some significance in explaining the spectral behavior in materials and LEDs. Many studies have been published (and also referenced in this manuscript) describing the spectral shape functions with only limited explanations on their use. So I cannot copy the application list from them either.
The main merit of the here described function is its pervasive nature. This means the same function can be applied for homogeneous and inhomogeneous spectra, possibly to evaluate material and structural properties or numerically simulate new devicec designs. Often the reader self knows best his reasons for interest and does not need a list for other possible applications.
For purely exponential spectral tails there are no reasons to include the homogeneous broadening and one should use only the Urbach function. However, in a few cases, it is necessary to include both broadening phenomena. 
There, it might enable the evaluation of electron and hole relaxation times and give valuable information about the disorder in the materials.
2) The sequence of the manuscript's results needs to be more obvious...
As this is mainly a comparison of available functions in the literature, I am reluctant to make the lineshape selections shorter and more obvious. It could result in the ditching of the hole manuscript as too obvious. The form of the main result, Eq. 13.  is already stated as inevitable (unquestionable, obvious) on the first line in sequence 6. The unique (less obvious) feature is mainly the picking of the included functions for Eq. 13.
3) The title of sequence '6. Discussion' has to change to '6. Conclusions'...
I agree with the reviewer and made this correction to the manuscript.

Reviewer 2 Report

The author considers an important subject from the point of view of the semiconductor laser theory. The origin and mechanisms of linewidth broadening determine the spectral and dynamical properties of devices. The discussion presented in the manuscript is of interest to the semiconductor laser society.

The discussion is well organized and the results are well presented. The author provides details concerning calculations (parameter values).

However, the manuscript references many fundamental books and papers, that can not be considered as recent works in the subject. I strongly recommend referencing recent papers and putting the work in contrast to their findings in order to enhance the perception of the results.

Overall, I recommend the manuscript for publication.

Author Response

English language and style:
Reviewer has suggested for spell checking:
After careful and concentrated read through the manuscript, I found two typing errors on the second line on page 7 (row no:164) '...ratio, it coverges to ..'  was corrected to '...ratio, it converges to....'.  Additionally, on the same page on the third line  (line no: 165) '... obeys the moment conservation...' was corrected to '...obeys the momentum conservation ...

Does the introduction provide sufficient background and include all relevant references? The reviewer comments that it can be improved.  
I agree that the literature references given in the introduction are not very recent. The reason is mainly the fact that quantum wells have not anymore for years been a hot research topic. In the recent journals, the focus has more or less shifted from them to quantum dots. Therefore, even the title of this manuscript might be despised as old-fashioned by many modern researchers. Personally I think that even the word 'quantum well' is an old-fashioned misnomer that should be changed to something like 'quantum trench' which would describe its shape much better. 

Reviewer 3 Report

The author presents a discussion of homogeneous and inhomogeneous broadening in quantum-well LEDs. It generalizes well known concepts of laser physics in a clear and pedagogic way, based on the most advanced and up to date understanding of solid-state physics.

I think the paper is worth of publication in Appl. Sci., after several minor corrections and clarifications are provided.

In general, different lineshapes are discussed. As for the Urbach tails, I am not sure I understand what criterion is used to say that the asymmetric lineshape is not reasonable. What is the "right" lineshape? I would provide a measured set of data or clarify the idea behing the conclusions.

Some style/linguistic points:

  1. "Pressure alias lifetime" not appropriate
  2. The use of decade is not clear. Does it mean a 1 difference in log scale? Normally 10log_10 is used in scientific literature, so a decade is 10 in this scale. Please, clarify.
  3. Line 76: (l. reduced) I do not understand
  4. Line 82, Exp must be in roman type
  5. After line 118, Eqs. 6 and 7 are commented before being shown.
  6. After line 154, the sentence "Taking the k-vecor [...]" is not very clear to me.
  7. In Fig. 3, the quantities in the legend are not explained
  8. Line 171, the use of parallel is not clear here
  9. Line 192, "emerging exponential integral function" I am not sure I understand what the author means here

Author Response

Thank you for the detailed and accurate comments and improvement suggestions! I greatly appreciate the careful and thorough reviewer.

The comments are covered in the following:

Is the research design appropriate? The reviewer answered: Can be improved.
I agree with this but without own experimental data, significant improvement cannot be reached easily. Now the conclusions are made after a literature review and convolution calculations.

Are the conclusions supported by the results? The reviewer answered: Can be improved.
1) In the review part of the manuscript, it was concluded that the lattice disorder produces both inhomogeneous and homogeneous broadening phenomena. In that sense, the resulting Eq. 13 in the manuscript is inevitable and does not require additional support. However, it is not argued that homogeneous contribution would be always observable and necessary in spectral analysis. In many cases, the Urbach tail is good enough.
2) What remains to be supported are the functions selected to Eq. 13.
-The selected homogeneous lineshape is correct with high certainty since only two physically explained functions could be found in the literature and the simple Lorentzian lineshape was found to fit poorly the experimental Urbach spectra in the literature. This result is depicted in Fig. 4. and explained in the adjacent text.
-Perhaps more difficult to convince the readers is the selected inhomogeneous Urbach lineshape. It was based assuming Gaussian distribution in the disorder potential. It is not written in the text since the readers already know that Gaussian distributions are symmetric, which is one argument against the asymmetric Urbach function. Another argument is the sharp peak that results from the convolution with the asymmetric Urbach function and is depicted in Fig. 2. This sharpness is never observed in experimental absorption or emission spectra. Fortunately, the symmetric and asymmetric Urbach lines convolve to similar low energy spectra in the bandgap.

Some Style/linguistic points:
1) on manuscript line 25: 'Pressure alias lifetime'  was changed to 'pressure-related lifetime' 
2) on several lines in the manuscript: word decade is corrected:
on line 101 text: 'may span several intensity decades' was changed to: 'may span several orders of magnitude',
on line 173 text: 'tail spans several intensity decades'  was changed to: 'tail spans several orders of intensity magnitude'.
on line 175 text: 'spectral tail may span several intensity decades' was changed to  'spectral tail may span several orders of magnitude'
on line 184 text: 'for two or more intensity decades' was changed to 'for two or more intensity magnitude orders'  
Reason: in the English language 'decade' means a time period of ten years and only in electrical engineering tenfold increase/decrease in frequency:  https://en.wikipedia.org/wiki/Decade_(log_scale)  
3) Line 76 . 'joint (l.reduced) density of states' was changed to 'joint density of states' 
Reason: While the 'joint density of states' and 'reduced density of states' are synonyms it is not necessary to express it here.
4) Line 82. 'Exp' was changed to 'Exp'.
Reason:  The Italic font comes from Latex typesetting for math functions but here the function is inside the bulk text.
5) After line 118 Eqs. 6 and 7 are commented before being shown. 
No change. These comments are right above the equations. There is no distraction possibility for the reader.
6) After line 154 the sentence, 'Taking the k-vector conservation..' 
This is a synonym of charge carrier (electron and hole) momentum conservation and is mentioned already in the introduction on line 65. Therefore, I do not want to make additions here, after line 154 unless the reviewer insists so. k-vector conservation is included in the unbroadened JDOS function, but not in the convolution of two Lorentzian functions on the energy scale. To include it in the broadening, it has to be examined separately, therefore Eq. 10.
7) In Fig. 3. the quantities in the legend are not explained. 
This was corrected by changing the caption to: 'Simulation for the homogeneous broadening in a qw-LED, using Eq. (10) and inhomogeneous line according to Eq. (8) with parameters: rhoJDOS ~H(E0), E0 = 1.4eV, Eb = Eu = 8meV \) and T = 293K' Now the parameters are the ones required in Eqs. 8. and 10.
8) Line 171. The use of parallel is not clear here. The sentence  was corrected to 'However, these two effects are simultaneously present at all times'
9) Line 192. 'instead of applying the emerging exponential integral functions' was changed to 'instead of applying the closed-form and exponential integral functions'.
Additionally in the caption of Fig. 4, Eg, Eb, and Eu were corrected to Eg Eb and Eu.